# Recurrent Gestational Diabetes Mellitus: A Narrative Review and Single-Center Experience

**DOI:** 10.3390/jcm10040569

**Published:** 2021-02-03

**Authors:** Aoife M. Egan, Elizabeth Ann L. Enninga, Layan Alrahmani, Amy L. Weaver, Michael P. Sarras, Rodrigo Ruano

**Affiliations:** 1Department of Endocrinology, Mayo Clinic, Rochester, MN 55905, USA; 2Department of Obstetrics and Gynecology, Mayo Clinic, Rochester, MN 55905, USA; Enninga.ElizabethAnn@mayo.edu (E.A.L.E.); layan.md@gmail.com (L.A.); ruano.rodrigo@mayo.edu (R.R.); 3Department of Obstetrics and Gynecology, Loyola University Medical Center, Maywood, IL 60153, USA; 4Department of Health Sciences Research, Mayo Clinic, Rochester, MN 55905, USA; weaver@mayo.edu; 5Department of Cell Biology and Anatomy, Rosalind Franklin University of Medicine and Science, Chicago, IL 60064, USA; michael.sarras@rosalindfranklin.edu

**Keywords:** gestational diabetes, GDM, recurrence, pregnancy, glucose intolerance

## Abstract

Gestational diabetes mellitus (GDM) is a frequently observed complication of pregnancy and is associated with an elevated risk of adverse maternal and neonatal outcomes. Many women with GDM will go on to have future pregnancies, and these pregnancies may or may not be affected by GDM. We conducted a literature search, and based on data from key studies retrieved during the search, we describe the epidemiology of GDM recurrence. This includes a summary of the observed clinical risk factors of increasing maternal age, weight, ethnicity, and requirement for insulin in the index pregnancy. We then present our data from Mayo Clinic (January 2013–December 2017) which identifies a GDM recurrence rate of 47.6%, and illustrates the relevance of population-based studies to clinical practice. Lastly, we examine the available evidence on strategies to prevent GDM recurrence, and note that more research is needed to evaluate the effect of interventions before, during and after pregnancy.

## 1. Introduction

Gestational diabetes mellitus (GDM) is defined as carbohydrate intolerance resulting in hyperglycemia of variable severity with onset or first recognition during pregnancy [1]. It excludes those who likely had overt diabetes prior to gestation, and it is one of the most common medical complications of pregnancy, with a prevalence of up to 30% depending on the population studied [2,3]. GDM develops when insulin synthesis and secretion are insufficient to overcome the physiologic insulin resistance that increases during all pregnancies. It is associated with an increased risk of adverse maternal outcomes such as pre-eclampsia [4] and cesarean delivery [5], and undesirable infant outcomes including macrosomia and neonatal hypoglycemia [6]. Although GDM tends to resolve after the pregnancy, up to 60% of affected women develop type 2 diabetes in the subsequent 10–20 years [7,8]. While causality has not been fully established, children born to women with GDM are more likely to have disorders of glucose metabolism and adiposity in later life [9,10].

Established risk factors for GDM include increasing body mass index (BMI), advanced maternal age, non-white ethnicity, and family history of diabetes mellitus [11,12]. Having GDM in a prior pregnancy is also accepted as a strong risk factor for GDM in a subsequent pregnancy [13]. Reports on GDM recurrence rates are highly variable and influenced by the GDM diagnostic criteria used as well as baseline population characteristics. For example, one meta-analysis of 18 cross-sectional studies revealed recurrence rates ranging from 29% to 80% [14]. In addition to clarifying the prevalence of GDM recurrence, an understanding of which women are at highest risk of GDM recurrence is of significant clinical importance. It will potentially allow targeted interventions to address modifiable risk factors and decrease the chance of recurrence, along with facilitating earlier assessment and intervention in a subsequent pregnancy.

The aim of this review is to outline the epidemiology of GDM recurrence, drawing on data from large population-based studies. We will then present single-center data from a tertiary hospital to illustrate how information from these larger studies can relate to individual clinical practice. Finally, we will discuss strategies that may reduce the risk of GDM recurrence.

## 2. Materials and Methods

### 2.1. Comprehensive Literature Review

Between May 2020 and September 2020, a literature search was conducted on PubMed and Embase. The following search terms were used alone and in combination: “gestational diabetes”, “GDM”, “recurrence”, “pregnancy”, and “glucose intolerance”. All studies published in the English language were considered, and a date restriction was not applied. Results were reviewed by the authors and selected for inclusion based on relevance to the topic. This was based on the judgment and agreement of the contributing authors. Additional articles were identified by manually searching reference lists of included articles.

### 2.2. Retrospective Study at Mayo Clinic

The retrospective cohort study included pregnant women who had their first pregnancy and were diagnosed with GDM between 1 January 2013 and 31 December 2017 at Mayo Clinic Rochester, Minnesota, U.S.A. It forms part of a series of studies aiming to better understand the molecular mechanisms underlying GDM and its relationship to future vascular complications [15]. The Mayo Clinic Investigational Review Board approved this retrospective study (Protocol: 17-009957). Searching the electronic medical records by the following ICD-9 and ICD-10 codes identified the cohort: 648.8, O24.410, O24.414, O24.419, O24.420, O24.424 and O24.343. Included women were between 18–45 years of age at the index pregnancy. Women with higher order pregnancies were excluded, as were women with established diabetes or a first trimester HbA1c >6.4%. This cohort was then followed until the end of 2018 to record whether or not they had a second pregnancy diagnosed with GDM.

The American College of Obstetricians and Gynecologists (ACOG) Clinical Management Practice Guidelines were used to diagnose GDM [16]. All women underwent a 50 g glucose challenge at 24–28 weeks gestation at Mayo Clinic (non-fasting). Following an overnight fast, women with a 1 h glucose of ≥140 mg/dL proceeded to a 3 h, 100 g oral glucose tolerance test. GDM was diagnosed if 2 or more of the following glucose thresholds were breached: fasting 95 mg/dL, 1 h 180 mg/dL, 2 h 155 mg/dL and 3 h 140 mg/dL.

A broad spectrum of clinical variables was collected for each pregnancy, including age, body mass index (BMI) at initial obstetric visit (<20 weeks in all cases), race, smoking history, family history, and GDM management. Abstracted data were collected in a secure online database. Comparisons between the two groups who did versus did not have GDM in their 2nd pregnancy were evaluated using the chi-squared test or Fisher’s exact test, as appropriate, for categorical variables and the two-sample *t*-test for continuous variables. All calculated *p*-values were two-sided.

## 3. Epidemiology of GDM Recurrence

The majority of GDM risk factors will persist or become worse in subsequent pregnancies; therefore, it is reasonable to anticipate a relatively high GDM recurrence rate. Table 1 summarizes key studies in the area of GDM recurrence. In 2007, Kim et al. published a systematic literature review including 13 studies—11 retrospective cohorts and two case-control studies [17]. The number of participants in the included studies ranged from 19 to 1322. In the retrospective studies, the rate of GDM recurrence ranged from 30% to 84%. In the case-control studies, the risk of recurrent GDM was calculated using an odds ratio (OR), comparing the odds of a previous GDM pregnancy in women with a current GDM pregnancy versus women with a normal current pregnancy. This OR was calculated at 15.0 and 23.0 for the two included studies [18,19]. One of the included studies (*n* = 651) noted a 2.4% risk of pre-gestational diabetes in the subsequent pregnancy [20]. One of the major difficulties in comparing these studies is the differing criteria used to diagnose GDM. However, despite these differences, those with predominantly non-Hispanic white populations had a recurrence rate of <40%. On the other hand, studies containing predominantly minority populations including African Americans, Latinas and Asians had recurrence rates of >50%. No other risk factors were consistently identified across the studies.

In 2016, Schwartz et al. conducted a further systematic review and found recurrence rates ranging from 30–80% [14]. They included 14 studies that examined risk factors for GDM recurrence, including two studies containing >1000 participants that were published subsequent to the aforementioned review by Kim et al. [17]. A meta-analysis was conducted in order to estimate the variability of risk factors for GDM recurrence and pooled effects. They found that increasing maternal age was a risk factor for GDM recurrence; however, the estimated difference in age between those with and without GDM recurrence was minimal at 1.32 years (95% confidence interval (CI) 0.89–1.76, *p* < 0.0001). Increasing maternal body mass index (BMI) was also identified as a risk factor with a weighted mean difference of 1.82 kg/m^2^ (0.39–3.26, *p* = 0.013). The authors examined the oral glucose tolerance test (OGTT) glucose concentrations in six studies that presented the mean and standard deviation of the fasting measurement for each group [14]. For post-glucose load measurements, they included four studies that used a 100 g, three-hour OGTT for diagnosis. Although the effect side was not large, there was an increased risk of GDM recurrence with increasing OGTT values, and the first three measurements had the largest impact (fasting glucose standardized mean difference (SMD) = 0.41, glucose one-hour post-load SMD = 0.32 and glucose two-hour post-load SMD = 0.41). Although the inter-pregnancy interval was not found to effect GDM recurrence, inter-pregnancy weight gain had a large effect (SMD = 0.78; *p* = 0.015). Use of insulin therapy (pooled OR 6.3 (95% CI 3.9–10.2), *p* < 0.001), multiparity (pooled OR 1.88 (95% CI 1.09–3.24), *p* < 0.02) and fetal macrosomia (pooled OR 1.63 (95% CI 1.25–2.13), *p* < 0.001) were also noted to increase the risk of GDM recurrence in a subsequent pregnancy [14].

A number of additional important publications were not included in these systematic reviews. Population-based data from Kaiser Permanente Southern California, USA. found that, compared to women without GDM in their first pregnancies (*n* = 65,132), women with a first pregnancy complicated by GDM (*n* = 2351) were at a significantly increased risk of GDM in their second pregnancy (OR 13.2, 95% CI 12.0–14.6) [21]. Overall, they reported a 52% recurrence rate of GDM. Compared to non-Hispanic white women, Hispanic women (OR 1.6, 95% CI 1.4–1.7) and Asian/Pacific Islanders (OR 2.1, 95% CI 1.8–2.3) had a higher GDM recurrence risk. This disparity was not explained when women were stratified according to inter-pregnancy interval.

A study from Kaiser Permanente, Northern California, including a total cohort of 22,351 women, of which 4.6% had GDM in the first pregnancy, noted that the age-adjusted risk of recurrent GDM in a second pregnancy was 38.19% (95% CI 34.9–41.42) [22]. This compared to a 3.52% (95% CI 3.27–3.76) risk in those whose first pregnancy was not complicated by GDM. In an analysis adjusted for multiple baseline variables, women with GDM in the first pregnancy had a 17-fold higher risk of developing GDM again in the second pregnancy (OR = 16.55 (95% CI 14.08–19.45)) as compared to those without GDM in their first pregnancy [22]. The risk of GDM recurrence did not differ between women with a BMI <25.0 kg/m^2^ and those with BMI ≥25.0 kg/m^2^. Pregnancy BMI was calculated in each pregnancy at the mean gestational age of 16.9 weeks. It was noted that women who had GDM in the first pregnancy, but not the second pregnancy, gained fewer BMI units between pregnancies than those who had GDM in both pregnancies, particularly those women with a BMI of ≥25.0 kg/m^2^ in the first pregnancy.

Another population-based study in the United States. included 4102 women with two sequential deliveries between 1998 and 2007, and GDM in the first pregnancy [23]. Depending on the method used to categorize maternal glucose status (birth certificate data versus hospital discharge records), the authors found a GDM recurrence rate of 34–48%. The authors also noted that estimates of progression from GDM in the first pregnancy to pregestational diabetes in the second pregnancy were 2.4–5.1%. Women with recurrent GDM were slightly older at the time of the first pregnancy than those who did not have GDM in the second pregnancy (29.7 versus 28.7 years, *p* < 0.001), and were more likely to have been born outside of the United States. (28.2 versus 25.3%, *p* < 0.05). BMI or weight gain data were not available for this study.

Outside of the United States, a retrospective review of women with GDM based in Australia noted a high rate of GDM recurrence (73.1%) [24]. Logistic regression analysis revealed pre-pregnancy BMI (aOR for BMI >30.0 kg/m^2^: 3.8 (95% CI 1.7–8.6) compared to BMI 18.5–25.0 kg/m^2^), fasting glucose concentration on OGTT (aOR for fasting glucose >5.2 mmol/L: 1.9 (95% CI 1.1–3.4) compared to ≤5.2 mmol/L), 2 h glucose concentration on OGTT (aOR for 2 h glucose >8.4 mmol/L: 2.7 (95% CI 1.6–4.7) compared to ≤8.4), and increasing categories of inter-pregnancy weight gain to be associated with recurrent GDM [24]. In fact, the OR for recurrent GDM among those who gained more than 8 kg was 20.5 (95% CI 5.0–84.5), compared with those who lost over 5 kg between the two pregnancies. In this cohort, recurrence of GDM was independent of ethnic backgrounds. Finally, a case control study based in China included 143 primiparous women who experienced GDM in their first pregnancy and went on to have a second pregnancy [25]. The authors observed a frequency of recurrent GDM of 55% and following adjustment for a number of factors, they identified the following risk factors for GDM recurrence: lower first trimester fasting plasma glucose (aOR 0.24 (95% CI 0.10–0.63)) and higher 1 h glucose (75 g OCTT) in the first pregnancy (aOR 1.43 (95% CI 1.09–1.87)) and higher first trimester triglyceride in the subsequent pregnancy (aOR 1.89 (95% CI 1.13–3.16)) [25].

## 4. Single-Center Experience of GDM Recurrence

### 4.1. Results

We identified 150 women who had their first delivery (live or still birth) between January 1, 2013, and December 31, 2017, and were diagnosed with GDM at Mayo Clinic. The majority of women were White (113/150, 75%), consistent with the background population in Rochester, Minnesota. Of these, 42 women had a subsequent delivery by the time of data collection, of which 20 (47.6%) were also diagnosed with GDM. The time between the two deliveries ranged from 0.8 to 3.6 years, with a mean of 1.9 (standard deviation 0.6) years.

Women with recurrent GDM were older at the time of the first pregnancy (mean age, 30.0 vs. 27.3 years; *p* = 0.03) (Table 2). A higher proportion of women with recurrent GDM required pharmacological therapy for GDM during the first pregnancy (45.0% (9/20) vs. 31.8% (7/22); *p* = 0.38) and a greater percentage of their first offspring had neonatal hypoglycemia (62.5% (10/16) vs. 36.8% (7/19); *p* = 0.13). However, these differences were not statistically significant. In addition, there was a tendency towards higher rates of pre-existing infertility in women who developed recurrent GDM (13.6% (3/22) vs. (35% (7/20); *p* = 0.15) (Table 3). Other variables including maternal weight, medical complications and mode of delivery were not associated with recurrent GDM (Table 2 and Table 3).

Figure 1 outlines the difference in maternal weight between the start of the first pregnancy and the start of the secondary pregnancy (inter-pregnancy weight gain). There was no difference between women with and without recurrent GDM.

### 4.2. Discussion

At Mayo Clinic, the GDM recurrence rate of 47.6% was in-keeping with the pooled recurrence reported in a previous meta-analysis [14]. Our cohort contained detailed individual patient data, and women with pre-existing diabetes were excluded. Within this cohort, older maternal age was a significant risk factor for GDM recurrence. In addition, a higher proportion of women with GDM recurrence required pharmacological therapy during their index pregnancy, and their infants were more likely to have had neonatal hypoglycemia. This suggests that women with more severe GDM may be at higher risk of recurrence in a subsequent pregnancy. It is interesting that a history of infertility is associated with GDM recurrence. This may relate to the fact that fertility challenges are more common in older women [26] and hormone therapies used during fertility treatments are associated with increases in insulin resistance [27]. In addition, the underlying cause of the infertility may be associated with GDM—for example, polycystic ovarian syndrome [28]. Additional predictors noted in prior studies were not observed in our cohort. This may be related to our relatively small sample size or differences in baseline population characteristics.

Large, population-based studies identify clinical characteristics associated with GDM recurrence; however, our data highlights the difficulty in applying this information to a clinical practice serving populations with different demographics, and women with several combinations of risk factors. Despite this, it would seem that GDM is more likely to occur if it was more severe in the index pregnancy, and in the setting of higher maternal age. While this information will help inform patient counselling, we recommend that all women with GDM are advised on their relatively high risk of recurrence in a subsequent pregnancy.

## 5. Prevention of Recurrent GDM

Many trials have examined the effect of various lifestyle interventions in the prenatal setting to prevent GDM. While they have typically included women with risk factors for GDM, they have not specifically targeted those with previous GDM. Some of these studies demonstrated modest effects on GDM prevention [29,30], but many more have shown no effect [31,32,33,34]. One exception in terms of target population was a 2016 randomized controlled trial by Guelfi et al. [35]. This trial involved a 14-week supervised, home-based, exercise program starting at 14 weeks of gestation. Unfortunately, while it was associated with important benefits for maternal fitness and psychological well-being, it did not prevent the recurrence of GDM [35]. The failure of this intervention, and indeed many of the GDM lifestyle prevention trials, is possibly due to their initiation at a relatively late stage in the pregnancy (typically the second trimester) and not taking into account patient heterogeneity including variations in BMI and insulin resistance [31]. The RADIEL study (Gestational Diabetes Mellitus Can Be Prevented by Lifestyle Intervention: The Finnish Gestational Diabetes Prevention Study), enriched by women with previous GDM, is another important study in this area [29]. This trial was based in Finland and recruited women <20 weeks gestation with a history of GDM and/or a pre-pregnancy BMI ≥25 kg/m^2^. Women were randomized to either a complex intervention (individual counseling on diet and physical activity), or standard antenatal care that included written information on diet and physical activity. The incidence of GDM was 13.9% in the intervention group versus 21.6% in the control group. The unadjusted analysis revealed this difference to be non-significant (*p* = 0.097), but it became significant following adjustment for a number of variables including age, pre-pregnancy BMI, previous GDM status, and number of weeks of gestation at time of oral glucose tolerance testing (*p* = 0.044). This study indicated that some women with previous GDM do respond favorably to a lifestyle intervention to reduce their risk of recurrence. We therefore await with interest the results of an ongoing randomized controlled trial in China, evaluating the efficacy of a dietary intervention in preventing the recurrence of GDM, with an estimated completion date of December 2021 [36]. In this study, women are enrolled in the first trimester of pregnancy. This approach is supported by a meta-analysis of 29 antenatal randomized controlled lifestyle trials by Song et al., involving 11,487 women [37]. The overall relative risk of GDM was 0.82 (95% CI 0.70–0.95), but this was due to a significant reduction in GDM among women where the intervention commenced at less than 15 weeks gestation (relative risk 0.78; 95% CI 0.64–0.96), with no significant reduction if the intervention commenced after this point (relative risk 0.97; 95% CI 0.82–1.13) [37].

There are no published trials that have evaluated pharmacological interventions to prevent recurrent GDM. The effect of metformin (initiated during pregnancy) on GDM prevention has been evaluated in two major trials, the 2015 EMPOWaR (effect of metformin on maternal and fetal outcomes in obese pregnant women) study [38] and the 2016 MOP (metformin versus placebo in obese pregnant women without diabetes mellitus) study [39]. Use of metformin for GDM prevention seems logical due to its positive effects on insulin sensitivity, lack of associated weight gain or hypoglycemia, and apparent safety in pregnancy. However, neither of these studies showed a reduction in the incidence of GDM. Data from the Diabetes Prevention Program Outcomes Study suggest that metformin can reduce progression to type 2 diabetes in women with a history of GDM and prediabetes [40]. However, in this study, there was an average interval of 12 years between the GDM pregnancy and initiation of metformin, and it remains unclear if metformin initiated postnatally in women with a history of GDM might lead to reduced risk of recurrent GDM. The TRIPOD (troglitazone in prevention of diabetes) [41] and the PIPOD (pioglitazone in prevention of diabetes) [42] studies demonstrated that thiazolidinediones may preserve β-cell function among women with a recent history of GDM. Although theoretically this may reduce the risk of recurrent GDM, this class of medications are not deemed safe in pregnancy and cannot be used in those who are actively planning to become pregnant. A double-blind, randomized controlled trial examining probiotics for the prevention of GDM in overweight and obese women found that this intervention was not effective, with similar rates of GDM in both arms of the trial (placebo 12.3% (25 of 204) versus 18.4% (38 of 207), *p* = 0.10) [43].

In those women who meet criteria based on BMI with/without medical comorbidities, the significant weight loss associated with bariatric surgery has been clearly linked with a reduction in risk of GDM [44,45]. As an illustrative example, Burke et al. examined 346 obese women who had a delivery before bariatric surgery (predominantly bypass procedures), and 354 who had a delivery after bariatric surgery. Women with a delivery after bariatric surgery had lower incidences of GDM (8% vs. 27%, OR 0.23, (95% CI 0.15–0.36)) [45]. While these studies tend not to focus on women with prior GDM specifically, one can assume that, in the appropriate clinical context, bariatric surgery will be associated with a reduction in risk of recurrent GDM.

Overall, there is a dearth of high-quality clinical evidence on how to reduce the risk of recurrent GDM. Until further information becomes available, it is reasonable to draw on the evidence from the broader GDM prevention studies, which suggest that intensive lifestyle interventions and significant inter-pregnancy weight loss may be effective in reducing risk. Indeed, women with a recent history of GDM may derive extra motivation from the potential of avoiding another high-risk pregnancy.

## 6. Conclusions

The GDM recurrence rate is high, with approximately 50% of women experiencing this same diagnosis in their subsequent pregnancy. Examination of large datasets suggests that the risk factors for GDM recurrence are similar to those for GDM itself, and include increasing maternal age, weight and certain ethnicities. In addition, women who require insulin to treat GDM in the index pregnancy seem more likely to experience recurrence. In practice, it is difficult to precisely determine an individual’s risk based on their clinical characteristics, and it is therefore reasonable to consider all women with previous GDM as being at high risk of recurrence. These women should receive appropriate counselling; weight loss is the intervention most likely to reduce recurrence risk; therefore, it should be supported when clinically indicated. Although not without challenges, future studies should seek to implement interventions before conception and extend through pregnancy and delivery to assess the maximum effect on GDM recurrence risk and pregnancy outcomes.

## Figures and Tables

**Figure 1 jcm-10-00569-f001:**
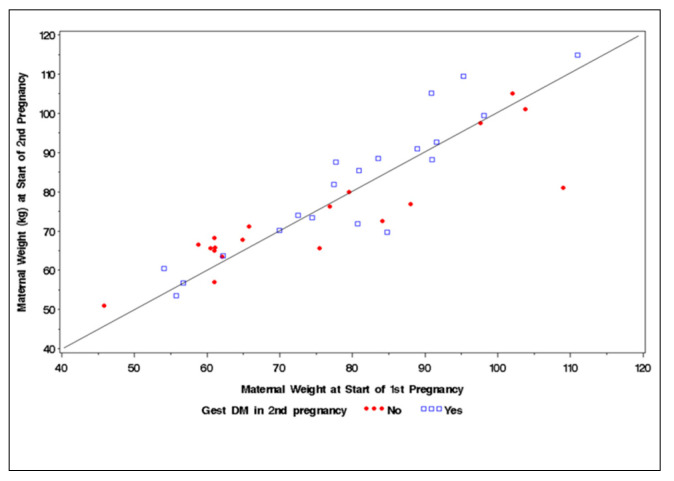
Scatter plot demonstrating interpregnancy weight change in women with and without recurrent gestational diabetes mellitus.

**Table 1 jcm-10-00569-t001:** Key cited papers: epidemiology of gestational diabetes mellitus recurrence with identified clinical risk factors (in index pregnancy unless otherwise specified).

**Meta-Analyses**	**N (with GDM in Index Pregnancy)**	**GDM Recurrence**	**Clinical Risk Factors for Recurrence**
Kim et al., 2007 [17]	13 studies—11 retrospective cohorts, 2 case-control studies; 3790 women	30–84% in retrospective cohorts; OR 15–23 for case control studies	Minority populations: African American, Latina, Asian
Schwartz et al., 2016 [14]	14 cross-sectional cohort studies; 9211 women	30–80%	Maternal age Maternal BMI Inter-pregnancy weight gain OGTT glucose concentrations Use of insulin Multiparity Fetal macrosomia
**Individual Studies**	**N (with GDM in Index Pregnancy), Location of Study**	**GDM Recurrence**	**Clinical Risk Factors for Recurrence**
Getahun et al., 2010 [21]	2351, USA	52%	Ethnicity: Hispanic & Asian/Pacific Islanders
Ehrlich et al., 2011 [22]	1028, USA	38%	Inter-pregnancy weight gain
England et al., 2015 [23]	4102, USA	34–48%	Maternal age Born outside of United States
Wong et al., 2019 [24]	3587, Australia	73%	Maternal BMI OGTT glucose concentrations Inter-pregnancy weight gain
Wang et al., 2019 [25]	143, China	55%	OGTT glucose concentrations First trimester triglycerides

BMI: body mass index; GDM: gestational diabetes mellitus; OGTT: oral glucose tolerance testing; OR: odds ratio; USA: United States of America.

**Table 2 jcm-10-00569-t002:** Maternal Characteristics.

Characteristic at Time of 1st Pregnancy	GDM in 2nd Pregnancy	
	No (*n* = 22)	Yes (*n* = 20)
Maternal race		
Asian	0 (0.0%)	1 (5.0%)
Black or African American	2 (9.1%)	0 (0.0%)
White	20 (90.9%)	18 (90.0%)
Unknown/Not Reported	0 (0.0%)	1 (5.0%)
Maternal Age (years), Mean (SD)	27.3 (3.8) *	30.0 (4.0) *
BMI (kg/m^2^) (<20/40 gestation), Mean (SD)	30.0 (8.8)	30.3 (5.9)
Current smoking	2 (9.1%)	1 (5.0%)
History of Infertility	3 (13.6%)	7 (35.0%)
First degree relatives with diabetes		
No	13 (59.1%)	19 (95.0%)
Yes	6 (27.3%)	0 (0.0%)
Unknown	3 (13.6%)	1 (5.0%)
**Characteristic at Time of 2nd Pregnancy**		
Maternal Age (years), Mean (SD)	29.1 (3.9)	32.1 (4.1)
BMI (kg/m^2^) (<20/40 gestation), Mean (SD)	29.8 (8.3)	30.2 (5.6)
Months between deliveries, Mean (SD)	20.9 (6.1)	24.2 (8.4)

Significant differences between groups are marked with * indicating *p* < 0.05.

**Table 3 jcm-10-00569-t003:** Pregnancy outcomes (first pregnancy).

Maternal Outcomes	GDM in 2nd Pregnancy	
	No (*n* = 22)	Yes (*n* = 20)
Gestational age at GDM diagnosis, Mean (SD)	28.4 (1.6)	27.0 (1.9)
Diabetes Management		
Diet and lifestyle modifications	15 (68.2%)	11 (55.0%)
Glyburide	7 (31.8%)	9 (45.0%)
Insulin	0	0
Pregnancy-induced hypertension	3 (13.6%)	4 (20.0%)
Pre-eclampsia	3 (13.6%)	2 (10.0%)
Type of Labor		
Spontaneous onset of labor at term	6 (27.3%)	11 (55.0%)
Induction	14 (63.6%)	8 (40.0%)
Preterm onset of labor (PTL)	2 (9.1%)	1 (5.0%)
Method of Delivery		
Vaginal Delivery	12 (54.5%)	14 (70.0%)
Cesarean	10 (45.5%)	6 (30.0%)
**Infant Outcomes**		
Birth Outcome		
Live Birth—Term (greater than 37 weeks gest.)	16 (72.7%)	15 (75.0%)
Live Birth—Preterm (less than 37 weeks gest.)	6 (27.3%)	4 (20.0%)
Stillbirth	0 (0.0%)	1 (5.0%)
Gestational age at delivery (weeks)		
Mean (SD)	38.5 (2.4)	38.8 (1.9)
Infants with research authorization	(*N* = 19)	(*N* = 16)
Admission to neonatal intensive care unit	0	1 (6.3%)
Admission to high dependency unit	9 (47.4%)	7 (43.8%)
Shoulder dystocia	1 (5.3%)	0
Neonatal hypoglycemia	7 (36.8%)	10 (62.5%)
Neonatal death	0	1 (6.3%)

There were no significant between group differences.

## Data Availability

All data for this study are available via Mayo’s RedCap system which can be accessed via email communication with the first author for an Excel Spreadsheet.

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
