# Peer review of "Recurrent Gestational Diabetes Mellitus: A Narrative Review and Single-Center Experience"

_jcm, 2021, doi:10.3390/jcm10040569_

Round 1

Reviewer 1 Report

This manuscript describes a review to find out the recurrence rate gestational diabetes in a next pregnancy. Actually, it finds out that it is not a systematic review, which disappointed me. Also, the data from the hospital of the authors is added as an example. Finally, strategies to prevent recurrence are mentioned. I think it is a little bit too much for one article.

The study population is rather small.

The English needs little improvement in grammar.

Title

Why add and clinical experience? What does it say? Try to think of something else or omit this phrase.

Why a narrative review while a systematic search is done?

Abstract

Complications are not common but frequently seen, rare or something like that.

The abstract only describes the plan but no results and conclusions.

The description of the search should be limited here.

Introduction

Good and concise.

Materials and methods

The literature review looks not properly described. This should be more elaborated. When was an article accepted for inclusion? How was the selection exactly done? If just a narrative review is done, it is not necessary in my opinion to elaborate or mention this search.

Which BMI was used in the hospital cohort? Preconceptionally or during the glucose tolerance test?

Epidemiology

The first part is described properly but it only describes the different systematic reviews so far. The only valuable part is the table with the results. I would have preferred a real systematic review. In the table of the individual studies also mention the countries.

Single center

Were the 150 women who were identified all the eligible women in this period? How were they selected?

Is it possible to add a control group and make a case-control study?

I miss a table with baseline characteristics of the 150 women.

Prevention

Well written and informative. I miss concluding remarks at the end of the paragraphs.

Conclusions

I miss more clearly which interventions women should use before conception. Make more clear that weight loss is the only way the recurrence rate can be lowered.

References

Good. Have a close look at the abbreviations used and if they are appropriate for this journal.

Reviewer 2 Report

Overall an excellent narrative review of the topic.

Are you able to make a comment about gestational weight gain for index & subsequent pregnancies especially in your own cohort of patients?  If you have this data available - can you please analyze this & add this to your paper?

The audience may be interested in this area - please make a note that use of probiotics did not reduce the risk of GDM as per the SPRING study published in 2019. 

Author Response

Thank you for your positive comments. Unfortunately we do not have data on gestational weight gain. Unlike inter-pregnancy weight gain, gestational weight gain has not been identified in previous studies as a risk factor for recurrent GDM, however, we do agree that it would be worthwhile evaluating this in future work in this area.

We have included the suggested reference to the SPRING study on page 9:

“A recent double-blind randomized controlled trial examining probiotics for the prevention of GDM in overweight and obese women found that this intervention was not effective, with similar rates of GDM in both arms of the trial [placebo 12.3% (25 of 204) versus 18.4% (38 of 207), p = 0.10] [43].”

Reviewer 3 Report

The manuscript "Recurrent Gestational Diabetes Mellitus: A Narrative Review and Clinical Experience" is original and well structured, gives important novelty to scientific literature. The review of literature is well performed and it considers all the potential useful scientific works to give a right conclusion to all the fields described in the text. Furthemore the statistical analyses is well conducted. The language is fine. It represents an interesting work and it gives the opportunity to focus the attention on the recurrence of gestational diabetes, to try to prevent it, and to  stimulate the scietific research on evaluation of the main risk factors related to it.

Author Response

Thank you for your positive comments. 

Round 2

Reviewer 1 Report

In my previous comments I gave significant options to improve this manuscript. I actually find that not much of my comments was implemented.So far, only very little was changed. 
